behaviour/cognition/psychology

associative learning, cognitive control, instrumental contingency, positive affect, reward

**Author for correspondence:**
Arthur Prével
e-mail: arthur.prevel@ugent.be

# Effect of non-instructed instrumental contingency of monetary reward and positive affect in a cognitive control task

Arthur Prével, Vincent Hoofs and Ruth M. Krebs

Department of Experimental Psychology, Ghent University, Henri Dunantlaan, 2, Ghent 9000, Belgium

AP, 0000-0001-5671-3155

In recent years, we observed a strong interest in the influence of motivation and emotion on cognitive control. Prior studies suggest that the instrumental contingency between a response and a rewarding or affective stimulus is particularly important in that context—which is resonating with observations in the associative learning literature. However, despite this overlap, and the relevance of non-instructed learning in real life, the vast majority of studies investigating motivation–cognition interactions use direct instructions to inform participants about the contingencies between responses and stimuli. Thus, there is little experimental insight regarding how humans detect non-instructed contingencies between their actions and motivational or affective outcomes, and how these learned contingencies come to influence cognitive control processes. In an attempt to close this gap, the goal of the present study was to test the effect of non-instructed contingent and non-contingent outcomes (i.e. monetary reward and positive affective stimuli) on cognitive control using the AX-continuous performance task (AX-CPT) paradigm. We found that entirely non-instructed contingencies between responses and positive outcomes (both monetary and affective ones) led to significant performance improvement. The present results open new perspectives for studying the influence of motivation and emotion on cognitive control at the insertion with associative learning.

# 1. Introduction

To successfully navigate in a rich and changing environment, humans rely on learned expectancies about positive and negative future outcomes, and on the ability to respond quickly to

conflicting events and changes in task demands. In psychology and neurosciences, these abilities are often summarized under the term cognitive control [1–4]. In recent years, a growing interest has been directed toward understanding how cognitive control interacts with motivation and emotion processes [5–8]. Motivation and emotion are known to be deeply related to goal-directed behaviours [9,10], and recent studies have investigated the impact of these processes on cognitive control. Prior studies employing motivational manipulations (using rewarding or punishing outcomes; e.g. [11]) and emotional manipulations (inducing positive or negative affect via pictures, music or videos; e.g. [12–14]) found performance modulation in cognitive control tasks. Specifically, motivational manipulations usually improve task performance, with shorter response time (RT) and/or higher accuracy. Interestingly, this beneficial effect of motivation has been reported across multiple cognitive control tasks and functions [11,15–18]. For example, evidence was found that reward increases the maintenance of cue information in working memory tasks, which is considered to reflect increased proactive control. Prominent demonstrations involve the AX-continuous performance task (AX-CPT) paradigm, in which participants have to respond to a probe stimulus (X) by executing a specific target response, but only when the probe is preceded a specific cue (A). Otherwise, participants have to produce a non-target response to less frequent sequences (typically, 70% of AX trials, and 10% of AY, BX and BY trials each). Increased maintenance of cue information caused by the presentation of reward is measured by global speed-up but decreased accuracy in AY trials [19], the performance decrement in AY trials being interpreted as an increased preparation to the target X from cue A. In contrast with this, embedding pictures or videos with positive or negative content in cognitive control tasks did yield mixed effects on performance. While studies reported more frequently increased cognitive flexibility [12,20] caused by (positive) affect, with lower error rates on AY trials and/or increased error rates on BX trials, other studies reported no impact of affective manipulations on cognitive control [21–23] or even increased cognitive stability/proactive control [14]. Overall, these observations are less homogeneous, contrasting with the results found in motivational experiments. Thus, even if motivation and emotion are related concepts, featuring overlap in the affective valence dimension [8,24–27], the above studies suggest a distinct influence of motivation and emotion on cognitive control functions.

Among the factors that are thought to influence the effect of motivation and emotion on cognitive control, the instrumental (or action–outcome) contingency between a response and a rewarding or affective stimulus seems to be particularly important. The global speeding and higher accuracy found with reward is observed in experiments in which reward presentation is contingent on correct (and usually fast) responding, and recent studies found that this effect decreases with non-contingent reward presentation, i.e. independent of the response performed by participants. For example, using an AX-CPT paradigm, Fröber & Dreisbach [20] only found performance improvement in comparison to a baseline (B) block when reward presentation was contingent (dependent on fast and correct performance), but not when it was non-contingent (see also [28,29]). In the emotional domain, most of the studies employed non-contingent manipulations, with affective stimuli being either presented as pre-cues [12] or after responses but independent of performance [23]. To the best of our knowledge, only one study by Braem et al. [30] tested the effect of contingent versus non-contingent presentation of affective pictures. The authors reported a distinct effect of the two conditions, with increased cognitive flexibility for contingent but not non-contingent positive pictures. This suggests that affective manipulations like motivational manipulations are influenced by the contingency between a response and a stimulus.

Overall, evidence for the role of instrumental contingency in cognitive control is in line with an extensive literature published within the associative learning domain, that has repeatedly demonstrated the influence of instrumental contingency on action frequency (for recent studies, see for example [31–33]). More exactly, it has been demonstrated that the frequency of an action increases when the action is consistently paired with (i.e. followed by) a positive value outcome (e.g. money, food, socio-emotional stimulus or aversive stimulus removal), but also that action frequency decreases when the response-outcome contingency is degraded, i.e. when the positive outcome is presented in the absence or independently of the action [34,35]. Note that with contingency degradation or non-contingent outcome (NCO), the stimulus can be presented either after responding (but independently of the response performed, e.g. correct or not), or at any moment during a trial (e.g. before target stimulus presentation), both resulting in performance degradation in comparison with contingent outcomes (COs). The effect of contingency is incorporated in several (if not all) contemporary models of instrumental associative learning (e.g. [36,37]), as well as in recent computational models of cognitive control (e.g. the expected value of control (EVC) model, [38]). More generally, the observed common role of instrumental contingency across action frequency and cognitive control fits

particularly well with an adaptive perspective of cognitive control, in which control functions are seen as influencing responses in order to obtain positive outcomes and avoid negatives ones, and with the recent proposition of grounding cognitive control research within associative learning [39–41].

Despite significant progress having been made in understanding how both motivation and emotion processing interact with cognitive control through different forms of contingency, it is notable that the vast majority of studies used direct instructions to inform participants about the contingencies between responses and stimuli (e.g. a correct and fast response will result in monetary reward (MR)). But in real life, cognitive control functioning often has to be adjusted without explicit instructions about the underlying contingency and is instead based on direct previous experience. Moreover, there is evidence from the associative learning literature that instructions enhance the effect of instrumental contingencies [42–44]—which might contribute to observed differences between motivation and emotion manipulations, in that the former more often feature explicit response-outcome contingencies. Together, there is little experimental insight regarding how humans detect non-instructed contingencies between their actions and motivational and affective outcomes, and how these learned contingencies come to influence cognitive control processes. This, however, seems highly relevant considering that real-life situations often lack explicit instructions. Using the AX-CPT paradigm, the present study explores the effect of non-instructed instrumental contingency of MR and positive affect (PA) on cognitive control. Inspired by the associative learning literature, our first aim was to test whether participants are able to detect and adjust their responses to these non-instructed contingencies (first research question). The contingent and non-contingent conditions were manipulated in discrete blocks as within-subject factors. In the contingent condition, positive outcomes were dependent on fast and accurate responses—unbeknown to the participant. To match the contingent condition, in the non-contingent condition, positive outcomes were delivered with the same probability, but independent of response speed and accuracy. In addition, we tested in how far the effect of non-instructed contingency differed between motivational and emotional outcomes (second research question), adding to previous work comparing different outcome types in the cognitive control domain. Specifically, motivational outcomes (i.e. objects signalling MR) and emotional outcomes (i.e. positive faces) were employed in two different groups of participants. Overall, according to the previous observations in the cognitive control and associative learning literature, we expected that contingencies between responses and outcomes are affecting performance in the AX-CPT paradigm, even in a context in which these are entirely non-instructed. More exactly, we expected that only a contingent presentation of a positive outcome would produce a significant RT decrease from a B level. In how far motivational and emotional outcome manipulations have a differential impact on cognitive control in this non-instructed contingency context remains an open question—considering partly inconsistent results in previous related studies.

# 2. Material and methods

## 2.1. Participants

A total of 108 participants took part in the study in exchange for 10 €. Participants in the MR group received an additional bonus of on average 5.6 €. Participants with empty cells in any of the phases by trial types conditions were excluded from the analysis (14 participants in total). Thus, the data of 94 participants were included in the analysis (48 in the MR group, 46 in the PA group, 76 females, $M_{age} = 22.34$, s.d. = 3.36, range = 18–35). The current sample size was based on related behavioural studies in the motivational and emotional domain (see [24,45]), and is consistent with studies using the AX-CPT paradigm cited above (e.g. [20]). Participants were recruited from the online recruiting of Ghent University. All participants were right-handed, with a normal colour perception, normal or corrected-to-normal vision and no reported history of diagnosed mental disorders. The experiment has been approved by the Ethical Commission at Ghent University, and a written informed consent was obtained from each participant before the experiment. Participants were informed of the testing time of approximately 60 min.

## 2.2. Stimuli and procedure

The experiment was programmed and presented using Matlab 2018b and the Psychophysics Toolbox [46]. The experiment was run on a desktop computer, with a monitor at a viewing distance of

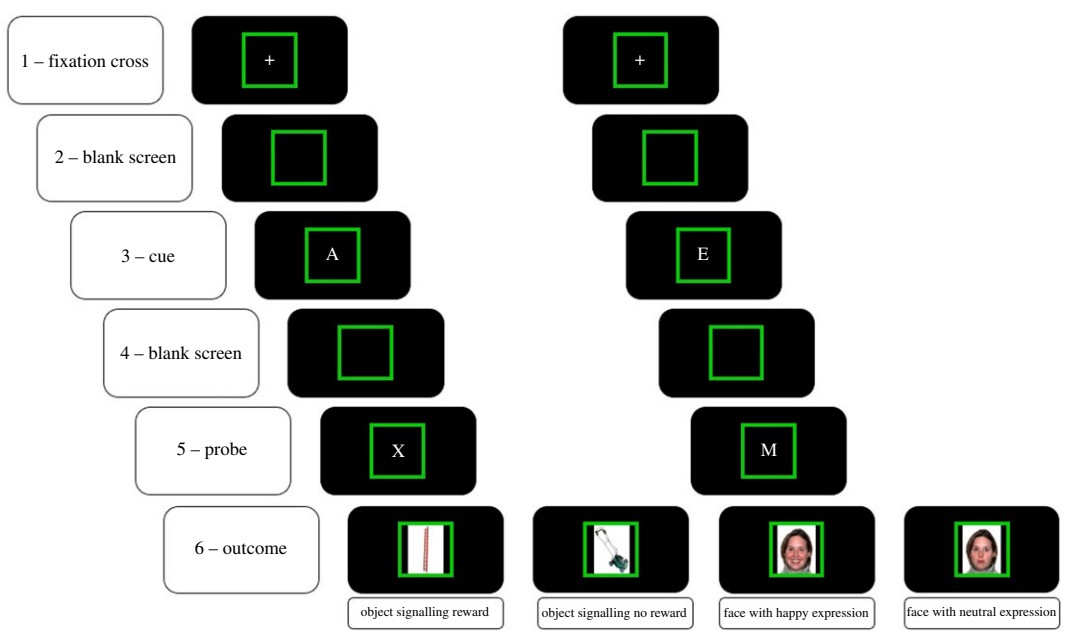

**Figure 1.** A trial started with (1) the presentation of a fixation cross for 200 ms, followed by (2) a variable blank screen between 2000 and 4000 ms. (3) The cue appeared for 300 ms followed by (4) a new blank screen of 1300 ms. (5) The probe was displayed for 1200 ms or until the participant responded. Finally, (6) responses were followed by either an outcome or a blank screen for 1200 ms, depending on the phase and the participant's performance. Trials were separated by a 1000 ms inter-trial interval. Coloured frames (here, green) signal different task phases.

approximately 80 cm (display resolution: $1920 \times 1080$ pixels). Responses were collected via a QWERTY keyboard with arrow keys '←' and '→' serving as the left and right response keys used in the experiment. Participants were free to position the keyboard at a comfortable position between themselves and the monitor. Similar to previous work in the emotion domain (e.g. [25]), faces with happy expression from the NimStim face stimulus set [47] were used as PA stimuli. Each face stimulus with happy expression was matched with a face stimulus with the same identity but neutral expression also taken from the NimStim, for a total of eight face stimuli (two females and two males, with both a positive and a neutral expression). In addition, eight object pictures chosen from the bank of standardized stimuli (BOSS) dataset [48] were used to signal the gain or absence of MR (figure 1 for an example of one set of object stimuli and one set of affective stimuli, and see the Appendix for the exact stimuli used in this study). Each trial of the AX-CPT consisted of a sequence of two letters presented centred on the screen (e.g. A followed by X) on a black background, with the first letter serving as a cue and the second letter serving as probe. The letters presented during the experiment consisted in either A, B, D, E, F or G, used as cues, and X, M, P, S, U or Z, used as probes (letter colour: white; letter font: default Matlab font; font size: 60). A target sequence consisted of the cue–probe sequence 'A followed by X' (i.e. AX trials) and required a target response (e.g. pressing the left arrow button). Non-target sequences consisted in any other cue–probe sequences (i.e. AY, BX or BY, with B representing any non-A cue, thus B, D, E, F or G, and Y representing any non-X probe, thus M, P, S, U and Z) and required a non-target response (e.g. pressing the right arrow button). The response mapping of the target and non-targets sequences to the left or right response key was randomized across participants. To induce a strong bias to the target response, and consistent with previous studies using the AX-CPT paradigm (e.g. [20]), AX trials occurred in 70% of trials, while AY, BX or BY trials occurred in 10% of trials each. Trial structure is illustrated in figure 1. Each trial started with the presentation of a white fixation cross for 200 ms, followed by a variable blank interval between 2000 and 4000 ms. Then, the cue appeared for 300 ms. After another blank interval of 1300 ms, the probe appeared and remained on the screen for 1200 ms or until the participant responded. Responses were followed by either an outcome or a blank screen (i.e. no outcome) for 1200 ms, depending on the ongoing contingency and the response performed (see details below). Trials were separated by a 1000 ms inter-trial interval. The experiment consisted of different phases (see below), which were associated with coloured frames (blue, RGB = [169, 234, 254]; yellow,

RGB = [239, 216, 7] or green, RGB = [0, 255, 0]; randomized across participants) in order to help participants to detect changes in the contingency structure without instructing them about the nature of the contingencies.

At the beginning of the experiment, participants were randomly assigned to the MR group or to the PA group. Participants were first instructed about the task and the response mapping (see the Appendix for detailed instructions) and received six practice trials of the AX-CPT paradigm (3 AX, 1 AY, 1 BX and 1 BY trials). During these practice trials, correct responses were followed by the word 'OK!'. Participants were then tested on 40 trials (28 AX, 4 AY, 4 BX and 4 BY trials, with trials order randomized) of a B phase, during which correct responses, incorrect responses and no responses, were followed by a blank screen. To assess the effect of contingency compared with B, contingent and non-contingent presentation of either object pictures signalling the gain of MR (MR group) or face pictures with a happy expression (PA group) were manipulated in two separated phases, i.e. CO phase and NCO phase. We elaborate on this below.

In the CO phase of the MR group, two object pictures taken from the BOSS dataset were randomly selected to signal the gain or absence of MR (figure 1). In the CO phase of the PA group, two face stimuli with the same face identity were randomly selected, i.e. one face with happy expression and one face with a neutral expression (figure 1). None of the face stimuli were associated with MR. The CO phase entailed 80 trials of the AX-CPT paradigm (56 AX, 8 AY, 8 BX and 8 BY trials, with trials order randomized). Correct and fast responses were followed by either the presentation of the object picture signalling the gain of MR (MR group) or the face stimulus with happy expression (PA group), while correct but slow responses were followed by either the second object picture signalling the absence of gain of MR (MR group) or the face stimulus with a neutral expression (PA group). Incorrect responses and misses were followed by a blank screen in both groups. The time threshold for correct and fast response was calculated and updated for each new trial (i.e. using percentile schedule method [49]). The threshold consisted in the 14th fastest response from the last 40 responses performed by a participant, and a new correct response had to be performed below that time threshold to be considered as fast (i.e. probability of getting a positive outcome: 35%). This method allowed to keep the rate of reward or PA stimulus (and hence positive outcome probability) constant across participants and across phases. Before the CO phase, participants in both groups were informed about the presentation of the different stimuli, but they were not instructed about any performance-outcome contingencies. In addition, participants in the MR group were also informed that each time the stimulus associated with MR was presented, they earned 0.10 €.

In the NCO phase of the MR group, two different object pictures were randomly selected to again signal the gain or absence of MR. Similarly, in the NCO phase of the PA group, two different face stimuli with the same identity were randomly selected, again one featuring a happy expression and one featuring a neutral expression, and none of these stimuli signalled the gain of MR. To implement a non-contingent presentation of these stimuli, a yoked design [50] of the CO phase was used (figure 2). Specifically, the 80 trials of the NCO phase (56 AX, 8 AY, 8 BX and 8 BY trials) were mirroring the trials order of the CO phase and the number of positive outcomes, ensuring a probability of getting a positive outcome of 35%, but these were presented randomly regardless of task performance (non-contingent). Importantly, to avoid phase order effects, half of the participants in both groups first performed the CO and then the NCO phase, and the other half received the reverse order. Because for the latter there was no CO phase to base the yoked procedure on, trial order and outcomes were based on a matched participant of the same group that performed the CO phase first. Here, the constant rate of positive outcomes across phases (NCO and CO) and participants was guaranteed by the use of the time threshold in the CO phase calculated and updated for each new trial, and maintaining a probability of getting a positive outcome at 35%. The presentation of reward or PA stimulus after responding in both the CO and the NCO phases allowed our procedure to test the unique effect of instrumental contingency on responding.

Finally, both after the CO and the NCO phases, participants were tested on 40 trials of the AX-CPT paradigm (28 AX, 4 AY, 4 BX and 4 BY trials, trials order randomized), during which correct responses, incorrect responses and no responses were always followed by a blank screen (extinction 1 (E1) and extinction 2 (E2)). These phases, which correspond to an instrumental extinction (i.e. outcomes are no longer delivered after responses), aimed to facilitate the discrimination between the CO and the NCO phases. In addition, we wanted to assess how changes in performances caused by contingent reward or affective stimuli persisted after the implementation of instrumental extinction. Thus, the final experimental design consisted of a B phase, a CO phase, a first extinction phase (E1), a NCO phase and a second extinction phase (E2). For half of the participants, the NCO phase and the E2 phase

| trail n | CO | | | NCO | | |
|---|---|---|---|---|---|---|
| | trail type | response | outcome | trail type | response | outcome |
| 1 | AX | slow | = | AX | fast | = |
| 2 | AX | slow | = | AX | fast | = |
| 3 | BY | fast | + | BY | incorrect | + |
| 4 | AX | slow | = | AX | slow | = |
| 5 | AY | incorrect | | AY | slow | |
| 6 | AX | fast | + | AX | slow | + |
| 7 | AX | fast | + | AX | slow | + |
| 8 | AX | incorrect | | AX | incorrect | |
| 9 | BX | fast | + | BX | slow | + |
| 10 | AX | slow | = | AX | fast | = |
| ... | ... | ... | ... | ... | ... | ... |

**Figure 2.** In the CO phase, correct and fast responses were followed by a positive outcome (+), i.e. either the presentation of the object picture signalling the gain of MR (MR group) or the face stimulus with happy expression (PA group). However, correct but slow responses were followed by a neutral outcome (=), i.e. either the second object picture signalling the absence of gain of MR (MR group) or the face stimulus with a neutral expression (PA group). Incorrect responses or absences of response were followed by a blank screen. In the NCO phase, the trials order and the outcomes received were similar to the CO phase, and outcomes were independent of the responses performed during this phase.

preceded the CO phase and the E1 phase. Again, each phase of the experiment was signalled to the participant by differently coloured frames. Note that the phases B, E1 and E2 were essentially the same (i.e. no outcomes delivered), and hence signalled by the same frame colour for each participant (again, colour-phase mapping was randomized across participants).

## 2.3. Data analysis

Analyses were performed using JASP v. 0.14.1 [51]. Raw and processed data and codes can be found on OSF [52]. RT (in milliseconds) and error rates (in percentages) served as dependent measures. Mean median RTs on correct responses and mean error rates were submitted to a $2 \times 4 \times 5$ mixed factors ANOVA, with outcome type (MR, PA) as between-subjects factor (i.e. group), and trial type (AX, AY, BX, BY) and phase (B, CO, E1, NCO, E2) as within-subject factors. Greenhouse–Geisser correction was applied where sphericity was violated. Significant main effects were followed up by Bonferroni-corrected *post hoc* comparisons. To increase confidence in the findings, frequentist mixed ANOVAs were followed by Bayesian mixed ANOVAs on the main effects and interactions with the same factors. These analyses were also performed using JASP v. 0.14.1. Inclusion Bayes factors ($BF_{inclusion}$) were used to assess the strength of evidence in favour of a particular effect. The r scales fixed effects, random effects and covariates were set at JASP default values, i.e. 0.5, 1.00 and 0.354, respectively. The primary focus of our analysis concerned the presence of an effect of phase on participants' performance as a result of the learned performance-outcome contingencies (first research question). We expected to find a significant main effect of Phase in RTs and significant performance benefits for the CO phase as compared with the other phases, as evidence that only contingent positive outcomes facilitate performance. We were also interested in an interaction between Phase and Trial Type, as an evidence of changes in the use of proactive control. Particularly, we expected an increased error rate on AY trials in the CO phase compared with the B, but not in the other phases. In addition, the second interest (second research question) concerned an effect of outcome type, as well as an interaction between outcome type and phase, to see how far motivational and emotional outcome manipulations have a differential impact on cognitive control.

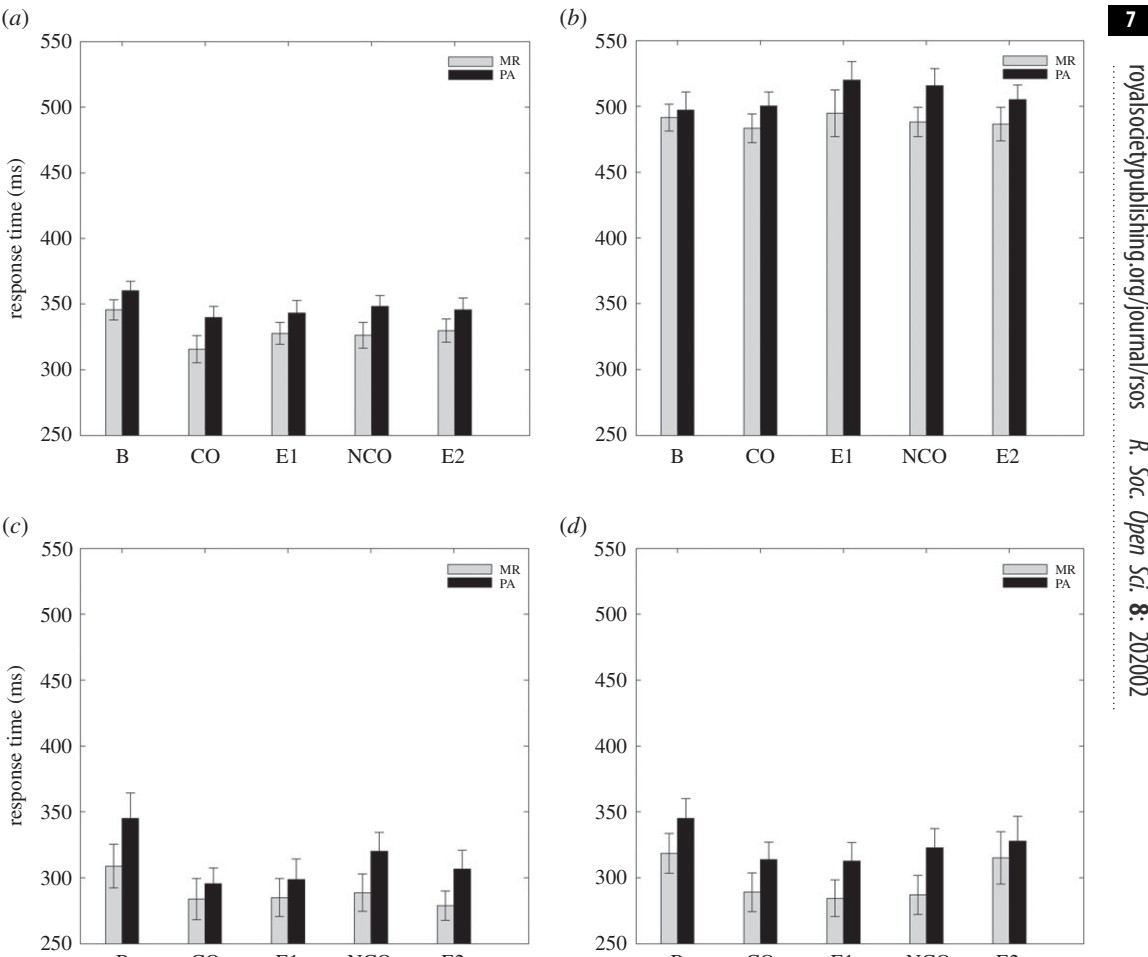

**Figure 3.** Mean RTs (ms) are plotted as a function of outcome type (MR, PA) and phase (B, CO, E1, NCO, E2) for each trial type (AX (a), AY (b), BX (c), BY (d)). Note that the order of phases CO and E1, and phases NCO and E2, was counterbalanced across participants so that B could be followed by CO and NCO.

## 3. Results

### 3.1. Response time results

Figure 3 shows the mean RTs as a function of outcome type (MR, PA) and phase (B, CO, E1, NCO, E2) for each trial type (AX, AY, BX, BY). Consistent with classic effects in the AX-CPT paradigm, the $2 \times 4 \times 5$ analysis revealed a significant main effect of trial type ($F_{2.189,201.362} = 498.315$, $p < 0.001$, $\eta_P^2 = 0.844$, $BF_{inclusion} = +\infty$), and *post hoc* testing showed a significant difference in RTs between all trial types (all $p < 0.001$), except between BX and BY ($p = 0.430$). Our data show the classic pattern of responses found in an AX-CPT paradigm, with longest RTs for AY trials and shortest RTs for BX and BY trials. The analysis also revealed a significant main effect of Phase ($F_{3.243,298.401} = 4.820$, $p = 0.002$, $\eta_P^2 = 0.050$, $BF_{inclusion} > 100$), and *post hoc* testing showed a significant decreased in RTs between B and CO ($p < 0.001$) but not between B and NCO ($p = 0.120$), which is consistent with our assumption that only contingent positive outcome would produce a beneficial effect on performance. However, the analysis also revealed a significant decrease in RTs between B and E1 ($p = 0.015$), suggesting a persistent effect of the contingent presentation of a positive outcome. The difference between B and E2 was not significant ($p = 0.105$). Finally, the analysis revealed no significant difference between CO and NCO ($p = 0.949$), nor between CO and E1 ($p = 1.000$) or between NCO and E2 ($p = 1.000$). The absence of significant difference between CO and NCO, or between CO and E1, were not expected, considering our assumption that only contingent positive outcome would show a beneficial effect on performance.

In addition, the frequentist analysis revealed a significant interaction between phase and trial type ($F_{6.887,633.611} = 2.864$, $p = 0.006$, $\eta_P^2 = 0.030$), with a simple main effect in AX ($F = 6.401$, $p < 0.001$), BX

($F = 5.483$, $p < 0.001$) and BY ($F = 3.568$, $p = 0.007$), but not in AY ($F = 0.979$, $p = 0.419$). This suggests that only the AX, BX and BY trials were influenced by contingencies manipulated in our experiment. However, *post hoc* testing conducted on AX trials shows no significant difference between B and CO, E1, NCO or E2 (all $p > 0.799$), and no significant difference between CO and E1 or NCO (all $p = 1.000$). The same analysis conducted on BX trials shows a significant difference between B and CO ($p = 0.005$) but also between B and E1 ($p = 0.014$), and B and E2 ($p = 0.020$). Comparisons between B and NCO, CO and E1, and CO and NCO were non-significant ($p = 1.000$). Finally, *post hoc* testing on BY trials show no significant difference between B and CO ($p = 0.112$), a significant difference between B and E1 ($p = 0.033$), and no significant differences for the other comparisons (all $p > 0.436$). Thus, the global main effect of Phase was not reflected at the trial type level. However, the Bayesian ANOVA on the Phase by Trial Type interaction revealed a moderate evidence in favour of the Null hypothesis ($BF_{inclusion} = 0.032$), which contrasts with the conclusion from the frequentist analysis and questions the reality of this significant interaction between phase and trial type. This difference between the two analyses will be discussed in the Discussion. Finally, there was no main effect of outcome type ($F_{1,92} = 2.699$, $p = 0.104$, $\eta_P^2 = 0.029$, $BF_{inclusion} = 0.692$) and no interaction between outcome type and phase ($F_{3.243,298.401} = 0.282$, $p = 0.853$, $\eta_P^2 = 0.003$, $BF_{inclusion} = 0.002$), between outcome type and trial type ($F_{2.189,201.362} = 0.202$, $p = 0.836$, $\eta_P^2 = 0.002$, $BF_{inclusion} = 0.006$), or between Outcome Type, Phase and Trial Type ($F_{6.887,633.611} = 0.576$, $p = 0.773$, $\eta_P^2 = 0.006$, $BF_{inclusion} < 0.001$). In summary, our results show no evidence of a differential effect on RTs between the presentation of MR and positive affective stimuli, which was not expected considering the differential effects frequently reported in the literature.

## 3.2. Error rate results

Figure 4 shows the mean error rates as a function of outcome type (MR, PA) and phase (B, CO, E1, NCO, E2) for each trial type (AX, AY, BX, BY). Like the results found with RTs and consistent with classic effects in the AX-CPT paradigm, the $2 \times 4 \times 5$ analysis revealed a significant main effect of trial type ($F_{1.764,162.254} = 71.603$, $p < 0.001$, $\eta_P^2 = 0.438$, $BF_{inclusion} > 100$). *Post hoc* testing showed a significant difference in error rates between all trial types (all $p < 0.005$) except between BX and BY ($p = 1.000$). There was no main effect of Phase ($F_{3.479,320.035} = 1.288$, $p = 0.277$, $\eta_P^2 = 0.014$, $BF_{inclusion} = 0.004$), but a significant interaction between phase and trial type ($F_{7.682,706.759} = 4.355$, $p < 0.001$, $\eta_P^2 = 0.045$, $BF_{inclusion} = 97.782$) with a simple main effect in AY ($F = 5.844$, $p < 0.001$), but not in AX ($F = 2.026$, $p = 0.090$), BX ($F = 0.325$, $p = 0.861$) or BY trials ($F = 1.486$, $p = 0.206$). *Post hoc* testing conducted on AY trials showed a significant difference with more errors between B and CO ($p < 0.001$), which is consistent with our assumption that contingent positive outcome would produce increased proactive control and more interferences in AY trials. However, the analysis revealed also a significant difference between B and E1 ($p < 0.001$) and between B and NCO ($p < 0.001$), and the difference was not significant between B and E2 ($p = 0.139$). Comparisons between CO and NCO, CO and E1, or between NCO and E2 were all non-significant ($p = 1.000$). Thus, even if error rates on AY trials were significantly higher compared with the B and numerically higher than in the other phases, the results show increased error rates also in the E1 and NCO phase. This result is consistent with what we found in the RTs analysis but is puzzling regarding our assumption that only contingent positive outcome would produce, here, significant increased errors in AY trials compared with the B phase.

Moreover, there was no main effect of outcome type ($F_{1,92} = 0.864$, $p = 0.355$, $\eta_P^2 = 0.009$, $BF_{inclusion} = 0.148$), but a significant trial type by outcome type interaction ($F_{1.764,162.254} = 3.999$, $p = 0.025$, $\eta_P^2 = 0.042$, $BF_{inclusion} = 66.061$) with a near significant simple main effect in AY ($F = 3.846$, $p = 0.053$), but not in AX ($F = 0.001$, $p = 0.974$), BX ($F = 1.702$, $p = 0.195$) or BY trials ($F = 0.353$, $p = 0.554$). Considering that numerically more errors were found in the MR group, this suggests a stronger interference caused by the presentation of MR than the presentation of positive affective stimuli. However, the interactions between phase and outcome type ($F_{3.479,320.035} = 1.226$, $p = 0.300$, $\eta_P^2 = 0.013$, $BF_{inclusion} = 0.013$) and between phase, outcome type and trial type ($F_{7.682,706.759} = 1.483$, $p = 0.163$, $\eta_P^2 = 0.016$, $BF_{inclusion} = 0.009$) were non-significant. This is consistent with the results found in RTs and suggests that the interference effect on AY trials is potentially stronger with MR than with PA, but the two outcome types do not differ in the nature of their effect on performance.

## 4. Discussion

Extensive research in cognitive sciences has demonstrated that instrumental learning about contingencies between our actions and their consequences is essential for navigating in complex and dynamic

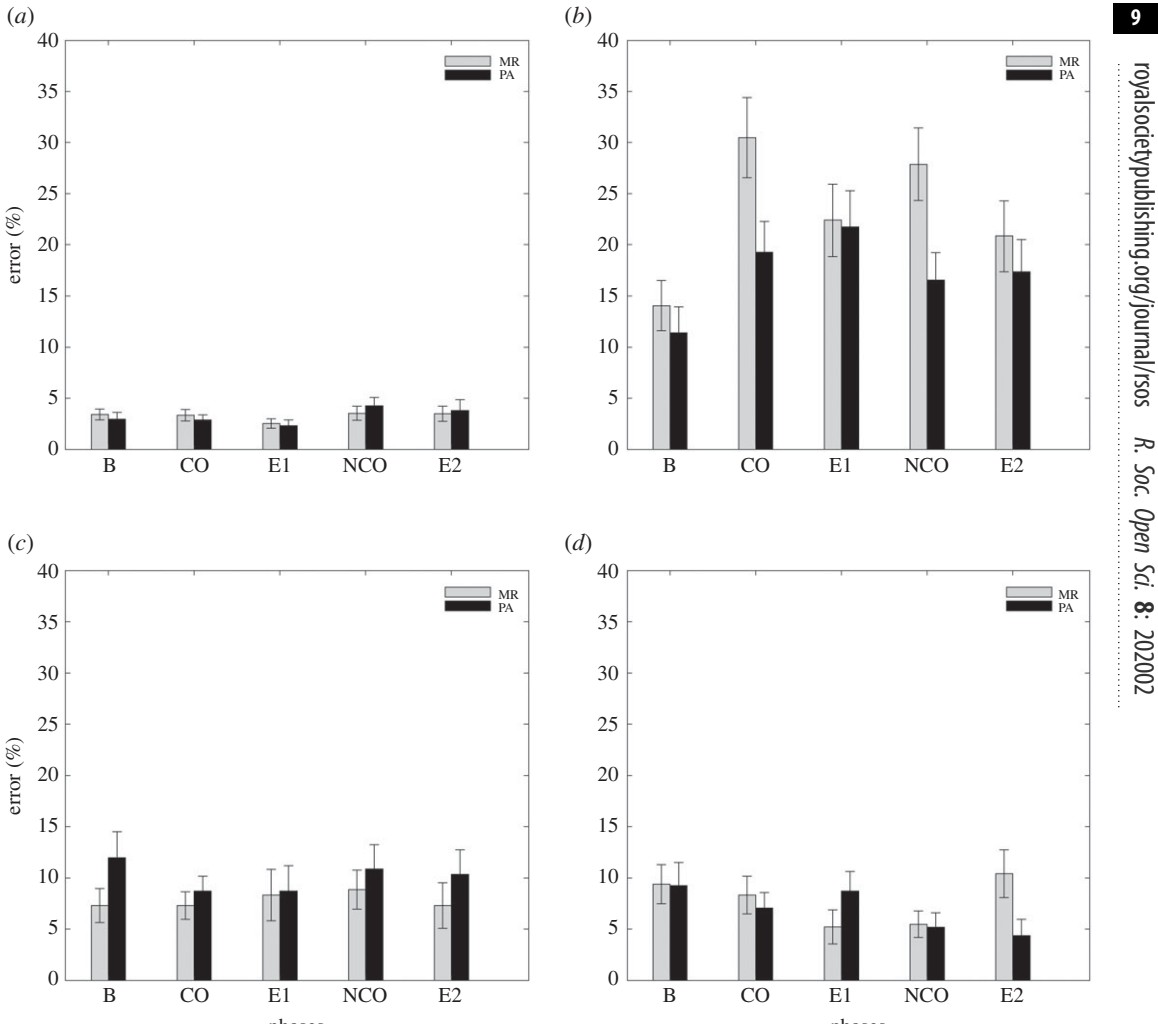

**Figure 4.** Mean error rates (%) are plotted as a function of outcome type (MR, PA) and phase (B, CO, E1, NCO, E2) for each trial type (AX (*a*), AY (*b*), BX (*c*), BY (*d*)). Note that the order of phases CO and E1, and phases NCO and E2, was counterbalanced across participants so that B could be followed by CO and NCO.

environments. This form of learning enables people to control the occurrence of events, i.e. to increase the probability of positive (appetitive) outcomes and to avoid negative (aversive) ones [9,10]. In this context, recent investigations have found an influence of instrumental contingencies with positive outcomes (MR or PA) on cognitive control [6–8]. But while in real life response-outcome contingencies are typically learned via non-instructed associations, laboratory studies using positive outcomes (and especially MR) to modulate cognitive task performance most often use explicit instructions. Here, we employ a non-instructed associative learning approach to explore the effect of non-instructed contingencies of MR and PA in the AX-CPT paradigm. The RT analysis revealed that only the introduction of contingent positive outcomes, but not non-contingent ones, resulted in a significant RT decrease compared with B performance. Interestingly, the RT analysis did neither reveal a significant difference between outcome types (group), nor a significant interaction between outcome type and phase or three-way interaction (outcome type, phase and trial type). Hence, when using similar non-instructed response-outcome contingencies, MR and PA seem to have similar effects on performance in a cognitive control task. The analysis of error rates revealed a significant trial type by phase interaction, as well as a simple main effect in AY trials, showing that response accuracy was decreased in this trial type dependent on changes in performance-outcome contingency. A significant interaction between outcome type and trial type, and a near significant simple main effect in AY trials suggest a lower accuracy in the MR group than in the PA group. However, the absence of an interaction between outcome type and phase (and three-way interaction) is in line with the RT data providing no evidence for a significant difference between motivational and emotional outcomes. The trial type by outcome

type interaction in error rate data might be interpreted as a stronger impact of MR as compared with PA on task performance due to greater behavioural relevance. Specifically, for an individual participating in a laboratory study, monetary incentives might be more relevant as compared with faces of unrelated people. However, this effect was independent of contingency, indicating that participants still learned to adjust their behaviour from PA outcomes.

Overall, our results resonate with previous findings in the cognitive control domain, which showed that MR can promote performance in general, and especially in trials probing proactive control. Importantly, by employing a non-instructed associative learning approach, we showed that participants were able to modulate their behaviour based on entirely non-instructed response-outcome contingencies, thus even in the absence of explicit instructions and/or cues (first research question), mirroring real-life behaviour in which prior explicit information regarding response-outcome contingencies is not always available. In line with previous studies [14,19], our experiment shows that contingent positive outcomes globally promote proactive control, as indexed by faster responses in an AX-CPT paradigm and more error in AY trials. Our experiment also replicates previous findings from Fröber & Dreisbach [20,28] demonstrating that only contingent (but not non-contingent) reward leads to response facilitation in comparison with B performance in the AX-CPT paradigm. Altogether, these results (including the present data) suggest that proactive control adjustments and performance modulations do not simply rely on the presence of positive outcomes, but seems to depend largely on the contingency between responses and outcomes, i.e. in how far participants can control the occurrence of outcomes. The present findings are conceptually consistent with recent adaptive accounts of cognitive control, according to which cognitive control functions influence the selection of responses based on the prospect of positive outcomes for engaging in those responses, or the avoidance of negative ones. According to the EVC model for example [38,53], a control signal is selected based on an (optimal) balanced between the costs and gains of selecting this control signal. Our data are consistent with this view in that in the non-contingent phases, engaging in more effortful cognitive control would have no effect on reward occurrence. In the future, it might be interesting to test the predictions of these models (for an alternative account to EVC, see for example [54]) for different levels of instrumental contingency.

Another important finding concerns the effect of non-instructed contingency with motivational and emotional outcomes. The behavioural modulation based on non-instructed contingencies was observed across different outcome types, with no evidence for a qualitative difference (i.e. same direction) between MR and positive effect (second research question). This latter finding is valuable in that previous related studies have reported partly inconsistent results. Contrasting with the consistent effect of MR (i.e. response speed-up and proactive control increment), PA manipulation has been classically associated with proactive control decrement [12,20], and sometimes small proactive control increment (compare with the effect of MR, [14]) or no effects [21–23]. Here, our results suggest that MR and PA stimuli might have globally a similar effect on proactive and task performance if they are manipulated in the same way. Thus, previous inconsistent findings might be the results of distinct contingency manipulations across motivational and emotional studies, i.e. accurate and fast responses usually required for (contingent) MR, versus affective stimuli presented independently of responses. As such, the results are also relevant with regard to previous contrasting findings in other task domains. For instance, different studies investigating conflict adaptation effects have shown that contingent reward outcomes promote conflict adaptation [16], while non-contingent affective outcomes abolish conflict adaptation [55]. Of course, this is not to say that reward and PA stimulus are completely exchangeable given similar contingency structures. Notably, neuroimaging studies have demonstrated overlapping neural responses across different positive valence stimuli, including primary and MRs, as well as emotional stimuli [25,56], which points to a common coding of basic valence. However, additional experimental features probably lead to a more differential processing, beyond mere valence (e.g. the motivational drive might be stronger for rewarding as compared with PA stimuli in laboratory studies). And in this context, the specific layout of the task and contingency structure will play a major role. Future experiments will be necessary to further explore similarities and differences between MR and PA stimuli in their effect on cognitive control, across different types of paradigms and instrumental contingencies. This approach would align with investigations on motivation and emotion in the associative learning domain, in which external stimuli are treated as outcomes or conditioned stimuli (i.e. stimuli predictive of outcomes), and their effect is both investigated at a behavioural (goal-directed behaviours, automatic responses) and neural level (see for example [57,58]).

Finally, when considering the above results in light of classic associative learning literature, two patterns are particularly intriguing. First, despite the observation that only the CO phase was

significantly different from B in RTs analysis, we found no significant difference between CO and NCO phases. Similarly, analysis of error rates shows a significant increase of errors in AY trials both in the CO and the NCO phases, and no significant differences between the two phases. This result is surprising considering that the presentation of MR or PA outcomes in the non-contingent phase was strictly independent of the response. However, the absence of a significant difference might be due to the task structure and to the yoked procedure used to implement a non-contingent presentation of the positive outcomes. Indeed, albeit independent of accuracy and speed, outcome presentation still followed responses directly in the non-contingent phase (which is different from purely random procedures in which outcomes can be presented at any moments during a trial, e.g. [59]). It is then possible that sometimes correct and fast responses were followed by a positive outcome, which may have produced a moderate performance improvement effect that might explain the absence of significant difference between the two phases. Another aspect of the task structure that might contribute to the absence of significant difference between the CO and the NCO phases is the adaptive RT threshold used in our experiment. The threshold for positive outcome presentation in the CO phase was calculated and updated for each new trial. While this guarantees equal reward probabilities across participants, results from a recent study conducted in our laboratory [60] show that such thresholding can have a detrimental effect on performance in the long run—potentially because independent of performance variation throughout the experiment, the reward frequency is kept constant. By contrast, a fixed threshold derived based on B performance may have a more persistent beneficial effect on performance. Thus, maybe the use of an adaptive threshold in the present experiment diminished the beneficial effect of contingent positive outcome and abolished a significant difference between the contingent and the non-contingent phases. To further explore non-instructed contingency effects, future studies may employ variations in the degree of contingency between the response to perform (e.g. a fast and correct response) and the positive outcome. In the associative learning domain, a common formalization of this contingency is: $\Delta P = P(o|a) - P(o|\sim a)$, where $o$ represents the outcome presented and $a$ represents the goal-directed action emitted [31–33,35]. Studies have demonstrated that the frequency of an action increases with $P(o|a) > 0$ and positive $\Delta P$ (i.e. positive contingency), while a performance decrement is found with $P(o|\sim a) > 0$, i.e. degraded or negative contingency. Further experiments might be dedicated to investigate how the effect of MR and PA on cognitive control changes with different values of $\Delta P$.

Second, we did not find a significant difference between the CO phase and the following extinction phase. This finding might seem surprising because it is at odds with a rich literature on how instrumental extinction results in the decrement of action execution and performance [61,62]. However, recent work by Hefer & Dreisbach [63–65] employing a rewarded AX-CPT paradigm shows that reward increased cue maintenance at the cost of decreased flexibility to new task/contingency conditions. Notably, the observations by Hefer & Dreisbach ([64], experiment 1) are particularly relevant in that increased cue maintenance persisted in a subsequent block even when reward was no longer available, which mirrors the present extinction phase. Concerning our data, here two hypotheses can be proposed to explain this finding. First, considering the highly variable outcome distribution within a phase (due to different trial types and the threshold update in each new trial maintaining a rate of positive outcome at 35%), it is reasonable to assume that the contingent phase involved partial reinforcement, and thus that the transition from CO to E1 involved a partial-reinforcement extinction effect (e.g. [66]), i.e. the evidence of a slower extinction process with intermittent or variable positive outcome than with continuous positive outcome. Another interpretation would be based on the assumption that the task acquired intrinsic reward value, driving participants to respond fast and correct even in the absence of external positive outcomes. It is also possible that the persistence of increased proactive control beyond the reward phase (similar to [64]) is one of the reasons for the absence of significant differences between the CO and the NCO phases, along with the contribution of the task structure suggested above. Specifically, for half of the participants, the NCO phase preceded the contingent one. It is hence possible that reward-triggered proactive control enhancement (partly) persisted during the NCO phase. And related to this, Hefer & Dreisbach [67] found that proactive control increases with time-on-task in an AX-CPT paradigm. Again, this could be another factor contribution to the absence of significant difference between the CO phase and the E1 or the NCO phases. To further illuminate the present observation, an interesting route for future studies may be to focus on how reward/PA-based performance modulations persist with varying instrumental contingency and lengths of learning phases. Thus, this would imply to investigate not only the direct effect of positive outcome on performance across different contingencies conditions, but also to investigate the long-term effect of these different contingencies. Overall, these two intriguing patterns confirm the importance for

future experiments of studying the effect of positive/negative outcomes in the context of non-instructed contingencies. Using non-instructed contingencies, we can investigate how humans detect new response-outcome contingencies in their environment, and how these learned contingencies cause an adjustment in control and decision-making processes. More exactly, we can chart the experimental characteristics that contribute to efficient response-outcome learning, i.e. faster and greater behavioural changes in cognitive control (and other) tasks. These investigations would echo a long tradition of research in the associative learning domain on the *determinants* of instrumental or Pavlovian/predictive learning [68–70]. Here, future investigations might be dedicated to test whether those determinants are the same for cognitive control processes and more 'basic' goal-directed or habitual responses.

Before the conclusion, we would like to discuss two additional issues. The first concerns a potential limitation related to the small trial number in the rare trial types (AY, BX, BY), which questions the stability of the findings, and particularly the interactions with trial type tested on RTs and error rates. For example, the analysis of the phase by trial type interaction on RTs resulted in inconsistent findings, with the frequentist ANOVA suggesting a significant interaction between the two factors, while the Bayesian ANOVA suggests a moderate evidence in favour of the Null hypothesis. Thus, even if it is the characteristics of the AX-CPT paradigm to have less observations on some of the trials, like for example on AY trials to produce strong interference, future studies will have to take this aspect into account to be sure that the study is well powered regarding interactions measured on trial types. Finally, we will discuss the recent findings in the light of potential underlying neural mechanisms and implications for the clinical domain. Concerning the neural processes, it is commonly established that the prefrontal cortex plays an important role in cognitive control, and notably in the active maintenance of cue information. Studies have demonstrated that reward prospect enhances prefrontal cortex activity, which in turn is thought to modulate performance in working memory tasks [71,72]. Moreover, a vast number of studies have shown that a network of cortical and subcortical regions (including the ventral striatum and the anterior cingulate cortex) are involved in increasing attention and cognitive control in various tasks to maximize reward outcomes [5,73]. More recently, the locus coeruleus-norepinephrine system has been linked to the allocation of cognitive control [74,75]. In the context of the present study, it might be interesting to investigate how far activity modulations in these regions would vary with the degree of contingency between the performed action and the outcome (e.g. with $\Delta'P'$), as well as with the type of outcome (MR versus positive affective stimuli). With regard to clinical implications, the evidence that performance in a cognitive control task is facilitated by positive outcomes, and that these modulations can persist even when the reward is no longer delivered, may be relevant for cognitive control training approaches. For example, the allocation of cognitive effort is reduced in patients with depression [76]. It could be interesting to investigate whether cognitive control training involving reward would increase the allocation of control in these patients, and in which conditions this increment could persist in time even in the absence of additional positive outcomes. That said, it is important to consider that not only the willingness to invest cognitive control is often impaired in depressed patients, but also the ability to process reward signals [77].

## 5. Conclusion

We believe that the results of the present study are relevant for a general understanding of how humans adapt to ad hoc changes in the environment, and in particular in response to relevant outcomes without prior instructions. The observation that non-instructed response-outcome contingencies can modulate performance in a cognitive control task resonate with the recent proposition of grounding cognitive control research within associative learning [39–41]. The current approach based on non-instructed learning may open new avenues for computational models of cognitive control as well as contingency manipulations for the training cognitive control functions in clinical settings.

Ethics. The experiment has been approved by the Ethical Commission at Ghent University, and a written informed consent was obtained from each participant before the experiment.

Data accessibility. Data and codes can be found on https://osf.io/qu5cb/?view_only=eaa17db45e4d401b999a2dd2 9f97a11a or with the doi:10.17605/OSF.IO/QU5CB.

Authors' contributions. A.P. and R.M.K. conceived and designed the experiment; A.P. programmed the task and wrote all the scripts; A.P. collected the data; A.P., V.H. and R.M.K. analysed the data; A.P., V.H. and R.M.K. wrote the manuscript. All authors approved the final version of the manuscript for submission.

Competing interests. The authors have no known conflicts of interest to disclose.

Funding. This study was supported by a starting grant of the European Research Council (ERC) under the Horizon 2020 framework (grant no. 636116 awarded to R.M.K.).

Acknowledgements. We would like to thank G. Dreisbach and four anonymous reviewers for their insightful comments on previous versions of this manuscript.

# Appendix

After an introductory message, participants received the following instructions on the screen:

'During the experiment, pairs of letters will be presented on the screen for multiple trials. A trial will consist in a fixation cross presented at the centre of the screen, followed after a short interval by a first letter (cue), and then a second letter (target). During a trial, if the letter A (cue) is followed by the letter X (target), you will have to press the left arrow button of the keyboard, while the X target stimulus is presented. For any other pairs of letters (for example: B followed by X or A followed by Y), you will have to press the right arrow button while the target is presented. During a trial, press the correct button only when the target is presented. A key pressing before the target will be considered as an incorrect trial'.

Then, after the practice trials, participants received the following instructions:

'The experiment is composed of different phases in which the stimuli presented change. Each phase will be signalled by a coloured frame. We will ask you to pay attention to the colour of the frames presented. During the experiment, breaks will be automatically proposed between each block of trials. You can freely decide when starting a new block of trials'.

Before the CO phase and the NCO phase, participants in the MR group received the following instructions:

'You are now going to another phase of the experiment. Pictures can be presented during the trials. Continue pressing the left (right) arrow button if the letter A is followed by the target X when the target is presented, or press the right (left) arrow button for any other combination. In the next phase, pictures will be regularly shown on the screen. Each time you see the picture on the right, and only if this image is shown, it means that you earned a bonus of 10 cents of Euros!'.

In the PA group, the same instructions were shown to participants, except with no mentioning about MR. Finally, before E1 and E2, participants received the following instructions:

'You are now going to another phase of the experiment. Continue pressing the left arrow button if the letter A is followed by the target X when the target is presented, or press the right arrow button for any other combination'.

Pictures taken from the BOSS dataset and used in the MR group are: Apron, Bowtie, Chair, Floorlamp, Ladder, Lawnmower, Lightswitch, Microphone. The pictures taken from the NimStim dataset and used in the PA group are 20M_HA_C, 20M_NE_C, 36M_HA_O, 36M_NE_C, 05F_HA_O, 05F_NE_C, 07F_HA_O, 07F_NE_C.

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
