## [Peer Review File · Royal Society Open Science]

Review History

RSOS-202002.R0 (Original submission)

Review form: Reviewer 1

Is the manuscript scientifically sound in its present form?

Yes

Are the interpretations and conclusions justified by the results?

Yes

Is the language acceptable?

Yes

Do you have any ethical concerns with this paper?

No

Have you any concerns about statistical analyses in this paper?

No

Recommendation?

Accept with minor revision (please list in comments)

Comments to the Author(s)

The authors of this manuscript aim to bridge largely separate literatures in to the effects of motivation and affect on cognitive control. They noticed – drawing on the literatures on implicit learning - how these have been studied in vastly different ways, where motivational incentives, such as reward, were mostly performance contingent, whereas affect manipulations were usually independent of performance. To bridge that gap, they manipulate the outcome type (monetary/no reward vs positive/neutral affective facial expression) and whether those are provided contingent on performance or not. Importantly, they yoke the overall probability of these outcomes across conditions, so that it is truly the contingency that varies and not the overall expectation.

This is a well-thought-out study that addresses an important question and bridges multiple extensive literatures that have touched remarkably little so far. They also employ a considerable sample size.

Unfortunately the results are not particularly impressive, but the authors do a great job discussing the limitations of their findings and outlining possible reasons and follow-ups. I believe this should not stand in the way of publication of this study.

I do think that it could be made a little easier for the reviewers to read this manuscript and appreciate the findings.

I have the following suggestions:

- 1) Reduce the use of acronyms OR if that's not possible provide a clear rationale for the acronyms that are used (e.g. don't leave it to the reader what PA stands for). Please also write the results in a way that readers can understand them without having read the method section. That is when there are acronyms, introduce them again. Better yet, don't use acronyms.
- 2) In the same vein: It is very difficult to follow the results – in particular in the current MS format with the Figures at the end. I would not require the reader to switch between text and figures. I believe it would help to walk the reader through the rationale – i.e. what is the relevant test – then provide the critical answer and then report what else was found and qualifies these findings. Please also spell out what the specific findings in the post hoc analyses are, so the reader doesn't need to refer to the figure.
- 3) For the results figures: I would recommend you reorder figures so that the relevant bars are next to each other. As I understand the relevant comparisons are between the phases within each condition, rather than between the conditions, as well as whether these effects differ between outcome types. That is really difficult to discern in the current figures. I think the comparison would be easiest if Trial Type and Outcome Type were swapped. That would also illustrate the trial type by phase interaction better.

Review form: Reviewer 2

Is the manuscript scientifically sound in its present form?

Yes

Are the interpretations and conclusions justified by the results?

No

Is the language acceptable?

Yes

Do you have any ethical concerns with this paper?

No

Have you any concerns about statistical analyses in this paper?

Yes

Recommendation?

Major revision is needed (please make suggestions in comments)

Comments to the Author(s)

I enjoyed reading the manuscript "Effect of implicit instrumental contingency of monetary reward and positive affect in a cognitive control task". The authors present a well-motivated study, the methods are sound, and the manuscript is well-written. Despite this overall positive evaluation, I have a few comments and concerns which prevent me from recommending publication of the manuscript in its present form.

- The authors provide a nice overview of the existing literature on reward and positive affect effects on cognitive control. But they missed a few relevant papers by Hefer and Dreisbach:
 - o Hefer & Dreisbach (2016). The motivational modulation of proactive control in a modified version of the AX-continuous performance task: Evidence from cue-based and prime-based preparation. *Motivation Science*, 2(2), 116-134. <http://dx.doi.org/10.1037/mot0000034>
 - o Hefer & Dreisbach (2017). How performance-contingent reward prospect modulates cognitive control: Increased cue maintenance at the cost of decreased flexibility. *JEP: LMC*, 43(10), 1643-1658. <http://dx.doi.org/10.1037/xlm0000397>
 - o Hefer & Dreisbach (2020). Prospect of performance-contingent reward distorts the action relevance of predictive context information. *JEP: LMC*, 46(2), 380-399. <http://dx.doi.org/10.1037/xlm0000727>
 - o Hefer & Dreisbach (2020). The volatile nature of positive affect effects: opposite effects of positive affect and time on task on proactive control. *Psychological research*, 84, 774-783. <https://doi.org/10.1007/s00426-018-1086-4>
- I'd like to highlight the findings from Hefer & Dreisbach's work that seem to be most relevant for the present study: 1) Hefer and Dreisbach found that the increased proactive control by reward comes at a cost of decreased flexibility in terms of delayed adaptation to new reward and task conditions. This seems to be important with respect to the comparisons with the extinction phases in the present study. Furthermore, there might be carry-over effects from the contingent reward phase to the non-contingent phase, if the latter is following the first. 2) Hefer and Dreisbach (as well as Fröber & Dreisbach, 2016) found time-on-task effects in the AX-CPT in terms of increasing proactive control over time. These time-on-task effects interfere with rather subtle (compared to reward effects) positive affect effects. Due to these two phenomena, I suggest to conduct an analysis with counterbalance order as an additional between-subjects factor. Collapsing across counterbalance order seems not justified, given the existing literature. Conclusions from the present study might be different when the additional factor is included in analyses.
- The study has a complex design and in my opinion, the results section would greatly benefit from a more detailed elaboration on the results. More precisely, the interaction of trial type and phase in RTs should be described in more detail than "there is a simple main effect in AX, BX, and BY, but not in AY". What exactly is meant by simple main effect here? I would have expected to find single comparisons between phases – separately for trial types – here, and not following the main effect of trial type. Likewise, the trial type x phase interaction in error rates is not described with enough detail.

Minor comment: Please add to the methods section, which exact stimuli from the NimStim face stimulus set and BOSS data set were used in the study.

Review form: Reviewer 3

Is the manuscript scientifically sound in its present form?

Yes

Are the interpretations and conclusions justified by the results?

Yes

Is the language acceptable?

Yes

Do you have any ethical concerns with this paper?

No

Have you any concerns about statistical analyses in this paper?

No

Recommendation?

Accept with minor revision (please list in comments)

Comments to the Author(s)

Prevel and colleagues present a behavioural study, investigating the effect of implicit instrumental contingency of monetary reward and positive affect on cognitive control using an impressively large number of participants. The authors explored whether participants are able to detect and adjust their responses to these implicit contingencies and tested how far the effect of implicit contingency differed between motivational and emotional outcomes. The authors found that implicit contingencies between responses and both monetary and affective outcomes led to significant performance improvement. Furthermore, both implicit reward domains impacted task performance similarly. This is a well-structured, well-written and methodologically sound behavioural study. The design is well-conceived and the statistical analysis appropriate. I fully recommend publication, however, I do request some additional discussion to widen the scope of this study.

Specifically, I would appreciate if the authors could add some more discussion regarding the potential underlying neural processes as well as some speculation as to how the results and or simply the interesting paradigm might be useful in the clinical domain.

Recently there have been several studies linking neural mechanisms of arousal and cognitive control to psychopathological disorders such as anxiety, depression or even to relapse after therapy.

Berwian IM, Wenzel JG, Collins AGE, Seifritz E, Stephan KE, Walter H, Huys QJM.

Computational Mechanisms of Effort and Reward Decisions in Patients With Depression and Their Association With Relapse After Antidepressant Discontinuation. *JAMA Psychiatry*. 2020 May 1;77(5):513-522. doi: 10.1001/jamapsychiatry.2019.4971. PMID: 32074255; PMCID: PMC7042923.

Grueschow M, Stenz N, Thörn H, Ehlert U, Breckwoldt J, Brodmann Maeder M, Exadaktylos AK, Bingisser R, Ruff CC, Kleim B. Real-world stress resilience is associated with the responsivity of the locus coeruleus. *Nat Commun*. 2021 Apr 15;12(1):2275. doi: 10.1038/s41467-021-22509-1. PMID: 33859187; PMCID: PMC8050280.

Grueschow M, Kleim B, Ruff CC. Role of the locus coeruleus arousal system in cognitive control. *J Neuroendocrinol.* 2020 Dec;32(12):e12890. doi: 10.1111/jne.12890. Epub 2020 Aug 20. PMID: 32820571.

Kaldewaij R, Koch SBJ, Hashemi MM, Zhang W, Klumpers F, Roelofs K. Anterior prefrontal brain activity during emotion control predicts resilience to post-traumatic stress symptoms. *Nat Hum Behav.* 2021 Feb 18. doi: 10.1038/s41562-021-01055-2. Epub ahead of print. PMID: 33603200.

I believe the manuscript could benefit from incorporating these papers and related work to outline potential paths for diagnosis, intervention or treatment related to psychopathological disorders and thereby substantially enhancing the scope of the results.

In the introduction the AX-CPT paradigm falls a bit out of the sky and leaves the non-expert reader wondering what this is. I believe it would be helpful to add one or two very general and explanatory sentences to alleviate this issue.

Is it possible that the following sentence: 'To match the contingent condition, positive outcomes were delivered with the same probability, but independent of response speed and accuracy.' Is missing a statement along the lines of 'in the non-contingent condition', just to make this clear? Page 5 line 56: 'Responses times' should read 'Response times'

Decision letter (RSOS-202002.R0)

Dear Dr Prével

The Editors assigned to your paper RSOS-202002 "Effect of implicit instrumental contingency of monetary reward and positive affect in a cognitive control task" have now received comments from reviewers and would like you to revise the paper in accordance with the reviewer comments and any comments from the Editors. Please note this decision does not guarantee eventual acceptance.

Please submit your revised manuscript and required files (see below) no later than 21 days from today's (ie 04-Jun-2021) date. Note: the ScholarOne system will 'lock' if submission of the revision is attempted 21 or more days after the deadline. If you do not think you will be able to meet this deadline please contact the editorial office immediately.

Please note article processing charges apply to papers accepted for publication in Royal Society Open Science (<https://royalsocietypublishing.org/rsos/charges>). Charges will also apply to papers transferred to the journal from other Royal Society Publishing journals, as well as papers submitted as part of our collaboration with the Royal Society of Chemistry

(<https://royalsocietypublishing.org/rsos/chemistry>). Fee waivers are available but must be requested when you submit your revision (<https://royalsocietypublishing.org/rsos/waivers>).

on behalf of Dr Inti Brazil (Associate Editor) and Essi Viding (Subject Editor)
openscience@royalsociety.org

Associate Editor Comments to Author (Dr Inti Brazil):

Comments to the Author:

Dear Dr. Prével,

First of all, thank you for your patience. The review process took a bit longer than expected, but I have managed to secure reviews from three experts in the field. As you can see, they have provided useful commentary that can guide you through the revision process, if you opt to revise the manuscript. In addition, I have concerns related to statistical rigor that require further clarification/justification if you choose to resubmit a revised version of the manuscript:

- 1) Please add a stronger (statistical) justification for the sample size. The argument that the N is similar to that used in prior studies is not convincing. This is particularly important given that you conducted a 2*4*5 mixed factors ANOVA, which is a relatively 'heavy' statistical design with a small N-per cell. I am concerned about the stability of the effects because of this;
- 2) The tone of the discussion should be adapted accordingly to reflect the above-mentioned issue and it should be discussed as a limitation;
- 3) One way to increase confidence in the stability of the findings would be to conduct alternative (more data-driven) analyses and showing that the results converge. For example, you could run Bayesian ANOVA, or even Permutation-based ANOVA (Kherad-Pajouh & Renaud, 2014). Especially the Bayesian ANOVA would allow you to test the relative strength of the interactions you are interested in, above and beyond the main effects. The results should be presented as an additional source of information. If the two approaches yield different results, you can discuss the implications for the conclusions that can be drawn from this particular study. If they converge, this will speak to the robustness of the findings.

Reviewer comments to Author:

Reviewer: 1

Comments to the Author(s)

The authors of this manuscript aim to bridge largely separate literatures in to the effects of motivation and affect on cognitive control. They noticed – drawing on the literatures on implicit learning - how these have been studied in vastly different ways, where motivational incentives, such as reward, were mostly performance contingent, whereas affect manipulations were usually independent of performance. To bridge that gap, they manipulate the outcome type (monetary/no reward vs positive/neutral affective facial expression) and whether those are provided contingent on performance or not. Importantly, they yoke the overall probability of these outcomes across conditions, so that it is truly the contingency that varies and not the overall expectation.

This is a well-thought-out study that addresses an important question and bridges multiple extensive literatures that have touched remarkably little so far. They also employ a considerable sample size.

Unfortunately the results are not particularly impressive, but the authors do a great job discussing the limitations of their findings and outlining possible reasons and follow-ups. I believe this should not stand in the way of publication of this study.

I do think that it could be made a little easier for the reviewers to read this manuscript and appreciate the findings.

I have the following suggestions:

- 1) Reduce the use of acronyms OR if that's not possible provide a clear rationale for the acronyms that are used (e.g. don't leave it to the reader what PA stands for). Please also write the results in a way that readers can understand them without having read the method section. That is when there are acronyms, introduce them again. Better yet, don't use acronyms.
- 2) In the same vein: It is very difficult to follow the results – in particular in the current MS format with the Figures at the end. I would not require the reader to switch between text and figures. I believe it would help to walk the reader through the rationale – i.e. what is the relevant test – then provide the critical answer and then report what else was found and qualifies these findings. Please also spell out what the specific findings in the post hoc analyses are, so the reader doesn't need to refer to the figure.
- 3) For the results figures: I would recommend you reorder figures so that the relevant bars are next to each other. As I understand the relevant comparisons are between the phases within each condition, rather than between the conditions, as well as whether these effects differ between outcome types. That is really difficult to discern in the current figures. I think the comparison would be easiest if Trial Type and Outcome Type were swapped. That would also illustrate the trial type by phase interaction better.

Reviewer: 2

Comments to the Author(s)

I enjoyed reading the manuscript "Effect of implicit instrumental contingency of monetary reward and positive affect in a cognitive control task". The authors present a well-motivated study, the methods are sound, and the manuscript is well-written. Despite this overall positive evaluation, I have a few comments and concerns which prevent me from recommending publication of the manuscript in its present form.

- The authors provide a nice overview of the existing literature on reward and positive affect effects on cognitive control. But they missed a few relevant papers by Hefer and Dreisbach:
 - o Hefer & Dreisbach (2016). The motivational modulation of proactive control in a modified version of the AX-continuous performance task: Evidence from cue-based and prime-based preparation. *Motivation Science*, 2(2), 116-134. <http://dx.doi.org/10.1037/mot0000034>
 - o Hefer & Dreisbach (2017). How performance-contingent reward prospect modulates cognitive control: Increased cue maintenance at the cost of decreased flexibility. *JEP: LMC*, 43(10), 1643-1658. <http://dx.doi.org/10.1037/xlm0000397>
 - o Hefer & Dreisbach (2020). Prospect of performance-contingent reward distorts the action relevance of predictive context information. *JEP: LMC*, 46(2), 380-399. <http://dx.doi.org/10.1037/xlm0000727>
 - o Hefer & Dreisbach (2020). The volatile nature of positive affect effects: opposite effects of positive affect and time on task on proactive control. *Psychological research*, 84, 774-783. <https://doi.org/10.1007/s00426-018-1086-4>

- I'd like to highlight the findings from Hefer & Dreisbach's work that seem to be most relevant for the present study: 1) Hefer and Dreisbach found that the increased proactive control by reward comes at a cost of decreased flexibility in terms of delayed adaptation to new reward and task conditions. This seems to be important with respect to the comparisons with the extinction phases in the present study. Furthermore, there might be carry-over effects from the contingent reward phase to the non-contingent phase, if the latter is following the first. 2) Hefer and Dreisbach (as well as Fröber & Dreisbach, 2016) found time-on-task effects in the AX-CPT in terms of increasing proactive control over time. These time-on-task effects interfere with rather subtle (compared to reward effects) positive affect effects. Due to these two phenomena, I suggest to conduct an analysis with counterbalance order as an additional between-subjects factor. Collapsing across counterbalance order seems not justified, given the existing literature. Conclusions from the present study might be different when the additional factor is included in analyses.

- The study has a complex design and in my opinion, the results section would greatly benefit from a more detailed elaboration on the results. More precisely, the interaction of trial type and phase in RTs should be described in more detail than "there is a simple main effect in AX, BX, and BY, but not in AY". What exactly is meant by simple main effect here? I would have expected to find single comparisons between phases – separately for trial types – here, and not following the main effect of trial type. Likewise, the trial type x phase interaction in error rates is not described with enough detail.

Minor comment: Please add to the methods section, which exact stimuli from the NimStim face stimulus set and BOSS data set were used in the study.

Reviewer: 3

Comments to the Author(s)

Prevel and colleagues present a behavioural study, investigating the effect of implicit instrumental contingency of monetary reward and positive affect on cognitive control using an impressively large number of participants. The authors explored whether participants are able to detect and adjust their responses to these implicit contingencies and tested how far the effect of implicit contingency differed between motivational and emotional outcomes. The authors found that implicit contingencies between responses and both monetary and affective outcomes led to significant performance improvement. Furthermore, both implicit reward domains impacted task performance similarly. This is a well-structured, well-written and methodologically sound behavioural study. The design is well-conceived and the statistical analysis appropriate. I fully recommend publication, however, I do request some additional discussion to widen the scope of this study.

Specifically, I would appreciate if the authors could add some more discussion regarding the potential underlying neural processes as well as some speculation as to how the results and or simply the interesting paradigm might be useful in the clinical domain.

Recently there have been several studies linking neural mechanisms of arousal and cognitive control to psychopathological disorders such as anxiety, depression or even to relapse after therapy.

Berwian IM, Wenzel JG, Collins AGE, Seifritz E, Stephan KE, Walter H, Huys QJM.

Computational Mechanisms of Effort and Reward Decisions in Patients With Depression and Their Association With Relapse After Antidepressant Discontinuation. *JAMA Psychiatry*. 2020 May 1;77(5):513-522. doi: 10.1001/jamapsychiatry.2019.4971. PMID: 32074255; PMCID: PMC7042923.

Grueschow M, Stenz N, Thörn H, Ehlert U, Breckwoldt J, Brodmann Maeder M, Exadaktylos AK, Bingisser R, Ruff CC, Kleim B. Real-world stress resilience is associated with the responsivity of the locus coeruleus. *Nat Commun*. 2021 Apr 15;12(1):2275. doi: 10.1038/s41467-021-22509-1. PMID: 33859187; PMCID: PMC8050280.

Grueschow M, Kleim B, Ruff CC. Role of the locus coeruleus arousal system in cognitive control. *J Neuroendocrinol.* 2020 Dec;32(12):e12890. doi: 10.1111/jne.12890. Epub 2020 Aug 20. PMID: 32820571.

Kaldewaij R, Koch SBJ, Hashemi MM, Zhang W, Klumpers F, Roelofs K. Anterior prefrontal brain activity during emotion control predicts resilience to post-traumatic stress symptoms. *Nat Hum Behav.* 2021 Feb 18. doi: 10.1038/s41562-021-01055-2. Epub ahead of print. PMID: 33603200.

I believe the manuscript could benefit from incorporating these papers and related work to outline potential paths for diagnosis, intervention or treatment related to psychopathological disorders and thereby substantially enhancing the scope of the results.

In the introduction the AX-CPT paradigm falls a bit out of the sky and leaves the non-expert reader wondering what this is. I believe it would be helpful to add one or two very general and explanatory sentences to alleviate this issue.

Is it possible that the following sentence: 'To match the contingent condition, positive outcomes were delivered with the same probability, but independent of response speed and accuracy.' Is missing a statement along the lines of 'in the non-contingent condition', just to make this clear? Page 5 line 56: 'Responses times' should read 'Response times'

===PREPARING YOUR MANUSCRIPT===

===PREPARING YOUR REVISION IN SCHOLARONE===

To revise your manuscript, log into <https://mc.manuscriptcentral.com/rsos> and enter your Author Centre - this may be accessed by clicking on "Author" in the dark toolbar at the top of the

page (just below the journal name). You will find your manuscript listed under "Manuscripts with Decisions". Under "Actions", click on "Create a Revision".

Author's Response to Decision Letter for (RSOS-202002.R0)

See Appendix A.

RSOS-202002.R1 (Revision)

Review form: Reviewer 1

Is the manuscript scientifically sound in its present form?

Yes

Are the interpretations and conclusions justified by the results?

Yes

Is the language acceptable?

Yes

Do you have any ethical concerns with this paper?

No

Have you any concerns about statistical analyses in this paper?

No

Recommendation?

Accept as is

Comments to the Author(s)

The authors have addressed all of my concerns. I have found the new results section much clearer and aligned with the rationale and it is much easier to discern the relevant effects in the revised figures. I particularly appreciate the authors' initiative to address points not raised by the reviewers, such as the implications of the online RT calibration - I think that is an important point and strengthened the manuscript.

I think the manuscript makes a valuable contribution to the literature and sheds light on an important variable that may help resolve some inconsistent previous findings and bridge research into the roles of motivation and affect in cognitive control.

Review form: Reviewer 2

Is the manuscript scientifically sound in its present form?

Yes

Are the interpretations and conclusions justified by the results?

Yes

Is the language acceptable?

Yes

Do you have any ethical concerns with this paper?

No

Have you any concerns about statistical analyses in this paper?

No

Recommendation?

Accept as is

Comments to the Author(s)

The authors did a great job in revising the manuscript. I have no further comments.

Review form: Reviewer 3

Is the manuscript scientifically sound in its present form?

Yes

Are the interpretations and conclusions justified by the results?

Yes

Is the language acceptable?

Yes

Do you have any ethical concerns with this paper?

No

Have you any concerns about statistical analyses in this paper?

No

Recommendation?

Accept as is

Comments to the Author(s)

All my comments were fully addressed. Well done.

Perhaps a final spell check would be good, i.e.: 'resent' should read 'recent'...

Decision letter (RSOS-202002.R1)

Dear Dr Prével,

It is a pleasure to accept your manuscript entitled "Effect of non-instructed instrumental contingency of monetary reward and positive affect in a cognitive control task" in its current form

for publication in Royal Society Open Science. The comments of the reviewer(s) who reviewed your manuscript are included at the foot of this letter.

You can expect to receive a proof of your article in the near future. Please contact the editorial office (openscience@royalsociety.org) and the production office (openscience_proofs@royalsociety.org) to let us know if you are likely to be away from e-mail contact – if you are going to be away, please nominate a co-author (if available) to manage the proofing process, and ensure they are copied into your email to the journal.

on behalf of Dr Inti Brazil (Associate Editor) and Essi Viding (Subject Editor)
openscience@royalsociety.org

Reviewer comments to Author:

Reviewer: 1
Comments to the Author(s)

The authors have addressed all of my concerns. I have found the new results section much clearer and aligned with the rationale and it is much easier to discern the relevant effects in the revised figures. I particularly appreciate the authors' initiative to address points not raised by the reviewers, such as the implications of the online RT calibration - I think that is an important point and strengthened the manuscript.

I think the manuscript makes a valuable contribution to the literature and sheds light on an important variable that may help resolve some inconsistent previous findings and bridge research into the roles of motivation and affect in cognitive control.

Reviewer: 2
Comments to the Author(s)

The authors did a great job in revising the manuscript. I have no further comments.

Reviewer: 3
Comments to the Author(s)

All my comments were fully addressed. Well done.
Perhaps a final spell check would be good, i.e.: 'resent' should read 'recent'...

Appendix A

Dr. Arthur Prével

Dr. Inti Brazil, Associate Editor
Royal Society Open Science

E arthur.aeac@gmail.com
T +32 9 264 64 07
M +33 6 036 612 00

Department of Experimental Psychology
Faculty of Psychology and Educational
Sciences
Henri Dunantlaan 2
9000 Gent
Belgium

DATE PAGE
09 July 2021 1/17

Manuscript Title: Effect of non-instructed instrumental contingency of monetary reward and positive affect in a cognitive control task - RSOS-202002

Dear Dr. Inti Brazil,

We hereby would like to submit the revised paper “Effect of non-instructed instrumental contingency of monetary reward and positive affect in a cognitive control task”, co-authored by Vincent Hoofs and Ruth M. Krebs to Royal Society Open Science. This is a revision of manuscript number RSOS-202002.

We are thankful for the opportunity to address the helpful comments by the editor and the reviewers. As a result, we believe that the manuscript has been improved. Below, we summarize our specific changes in response to the editor’s and reviewers’ comments.

Sincerely,

Arthur Prével, on behalf of all co-authors

DATE
09 July 2021PAGE
2/17

In what follows, we address the individual remarks from each reviewer. Italic text identifies the reviewers' comments and roman text identifies our response (with new text underscored). Changes to our manuscript are cited throughout this response letter and are highlighted in the revised manuscript file by colored font. We would like to emphasize that we changed the term "implicit" to "non-instructed", both in the title and within the manuscript. The reason we made this change is because we did not test in our experiment whether participants had a conscious representation of the (non-)contingent positive outcomes in our task. Thus, the term "non-instructed" seemed more appropriate in this context. Finally, we would like to highlight also that we added a short paragraph in the discussion introducing an additional factor that might contribute to the absence of a significant difference between the Contingent Outcome and the Non-Contingent Outcome phases (page 7). This is based on a recent study from our lab (Prével et al., 2021) that demonstrated a detrimental effect of the adaptive response-time threshold used in the present study.

Editor

*1) Please add a stronger (statistical) justification for the sample size. The argument that the N is similar to that used in prior studies is not convincing. This is particularly important given that you conducted a 2*4*5 mixed factors ANOVA, which is a relatively 'heavy' statistical design with a small N-per cell. I am concerned about the stability of the effects because of this;*

We understand your concern about the small N-per cell for some of the trial types manipulated in this experiment, and this is indeed an important point concerning the stability of the effects measured in our study. This small N-

DATE
09 July 2021PAGE
3/17

per cell for some of the trial type is inherent to the AX-CPT procedure we used. The AX-CPT procedure includes much more observations on AX trials than on the others trials (AY, BX, BY) in order to induce a bias in responding. Increasing the number of N-per cell on these trials would have imply to increase dramatically the global number of trials in our task, which was not possible considering the time constraint for testing. Thus, we decided to have a sample size consistent with previous studies within the field of cognitive control and reward manipulation, combined with a relative high number of trials per phase, in order to reach at least 1600 observations per phase-condition (Brysbaert & Stevens, 2018), which was the main focus of our study. However, we agree that the small N-per cell in the AY, BX, and BY trials is a limitation and have included a comment in the discussion of the revised manuscript (see below). In addition, following your recommendation, we ran an additional Bayesian ANOVA on main effect and interactions to test the stability of the findings. The results of this additional ANOVA are reported in the manuscript and discussed below in our letter.

2) The tone of the discussion should be adapted accordingly to reflect the above-mentioned issue and it should be discussed as a limitation;

We have included the considerations regarding the small N-per cell on some of the conditions in the revised discussion. More exactly, in the last paragraph we discussed the stability of our findings (page 8): “Before the conclusion, we would like to discuss two additional issues. The first concerns a potential limitation related to the small trial number in the rare

DATE
09 July 2021PAGE
4/17

trial types (AY, BX, BY), which questions the stability of the findings, and particularly the interactions with trial type tested on RTs and error rates. For example, the analysis of the phase by trial type interaction on RTs resulted in inconsistent findings, with the frequentist ANOVA suggesting a significant interaction between the two factors, while the Bayesian ANOVA suggests a moderate evidence in favour of the Null hypothesis. Thus, even if it is the characteristics of the AX-CPT paradigm to have less observations on some of the trials, like for example on AY trials to produce strong interference, future studies will have to take this aspect into account to be sure that the study is well powered regarding interactions measured on trial types.

3) One way to increase confidence in the stability of the findings would be to conduct alternative (more data-driven) analyses and showing that the results converge. For example, you could run Bayesian ANOVA, or even Permutation-based ANOVA (Kherad-Pajouh & Renaud, 2014). Especially the Bayesian ANOVA would allow you to test the relative strength of the interactions you are interested in, above and beyond the main effects. The results should be presented as an additional source of information. If the two approaches yield different results, you can discuss the implications for the conclusions that can be drawn from this particular study. If they converge, this will speak to the robustness of the findings.

As mentioned above, in addition to the classic frequentist ANOVA we ran a Bayesian ANOVA on main effect and interactions as an additional source of information. These analyses were performed on RTs and error rates, and the results of the analyses are included in the results section (pages 4-6).

DATE
09 July 2021PAGE
5/17

Overall, the analyses are consistent with the conclusion from the frequentist ANOVA, and particularly concerning the main effect of Phase on RTs and the absence of significant interaction between Phase and Outcome Type. However, the analysis did not reveal a Trial Type by Phase interaction on RTs, which is inconsistent with the results from the frequentist analysis. We would like to thank you for your suggestion of running an additional Bayesian ANOVA, which allowed us to qualify our findings. This divergence in the analyses is mentioned in the results section and discussed in the discussion. However, we do not think that this discredits the main observation from our study, that is, the observation of a positive effect on performance from contingent positive outcome, even in the absence of direct instructions. A result that is supported by both the frequentist ANOVA and the Bayesian ANOVA.

Lastly, for transparency we would like to highlight that we conducted the Bayesian ANOVA with the latest version of JASP (i.e., 0.14.1). Because the initial frequentist ANOVAs were made with a previous version, we decided to run again the whole analysis using the last version of JASP. Analyses of main effects, interactions, and simple main effects show similar results, but the post hoc testing on phases (RTs) show slightly different p values. Overall, this does not change the conclusions from our study, but for transparency we wanted to highlight this slight change in the RTs results.

Reviewer: 1

The authors of this manuscript aim to bridge largely separate literatures in to the effects of motivation and affect on cognitive control. They noticed – drawing on the

DATE
09 July 2021

PAGE
6/17

literatures on implicit learning - how these have been studied in vastly different ways, where motivational incentives, such as reward, were mostly performance contingent, whereas affect manipulations were usually independent of performance. To bridge that gap, they manipulate the outcome type (monetary/no reward vs positive/neutral affective facial expression) and whether those are provided contingent on performance or not. Importantly, they yoke the overall probability of these outcomes across conditions, so that it is truly the contingency that varies and not the overall expectation.

This is a well-thought-out study that addresses an important question and bridges multiple extensive literatures that have touched remarkably little so far. They also employ a considerable sample size.

Unfortunately the results are not particularly impressive, but the authors do a great job discussing the limitations of their findings and outlining possible reasons and follow-ups. I believe this should not stand in the way of publication of this study.

I do think that it could be made a little easier for the reviewers to read this manuscript and appreciate the findings. I have the following suggestions:

We are thankful for the Reviewer's very helpful suggestions on the previous version of our manuscript, and for the overall positive evaluation.

1) Reduce the use of acronyms OR if that's not possible provide a clear rationale for the acronyms that are used (e.g. don't leave it to the reader what PA stands for). Please also write the results in a way that readers can understand them without having read the method section. That is when there are acronyms, introduce them again. Better yet, don't use acronyms.

DATE
09 July 2021

PAGE
7/17

Thank you for this suggestion. We agree that acronyms do not promote accessibility of the manuscript and have removed them.

2) In the same vain: It is very difficult to follow the results – in particular in the current MS format with the Figures at the end. I would not require the reader to switch between text and figures. I believe it would help to walk the reader through the rationale – i.e. what is the relevant test – then provide the critical answer and then report what else was found and qualifies these findings. Please also spell out what the specific findings in the post hoc analyses are, so the reader doesn't need to refer to the figure.

We thank the reviewer for this suggestion. We now elaborate more on the relevant tests in the revised Data Analysis section (page 4): “The primary focus of our analysis concerned the presence of an effect of Phase on participants’ performance as a result of the learned performance-outcome contingencies (1st research question). We expected to find a significant main effect of Phase in RTs, and significant performance benefits for the Contingent Outcome phase as compared to the other phases, as evidence that only contingent positive outcomes facilitate performance. We were also interested in an interaction between Phase and Trial Type, as an evidence of changes in use of proactive control. Particularly, we expected an increased error rate on AY trials in the Contingent Outcome phase compared to the baseline, but not in the other phases. In addition, the second interest (2nd research question) concerned an effect of Outcome Type, as well as an interaction between Outcome Type and Phase, to see how far motivational

DATE
09 July 2021

PAGE
8/17

and emotional outcome manipulations have a differential impact on cognitive control.” In addition, in the Results section we described our findings in more detail and discussed their (in)consistency with our expectations.

3) For the results figures: I would recommend you reorder figures so that the relevant bars are next to each other. As I understand the relevant comparisons are between the phases within each condition, rather than between the conditions, as well as whether these effect differ between outcome types. That is really difficult to discern in the current figures. I think the comparison would be easiest if Trial Type and Outcome Type were swapped. That would also illustrate the trial type by phase interaction better.

This is a very interesting suggestion. Trial Type and Outcome Type are now swapped in the figures 3 and 4.

Reviewer: 2

I enjoyed reading the manuscript “Effect of implicit instrumental contingency of monetary reward and positive affect in a cognitive control task”. The authors present a well-motivated study, the methods are sound, and the manuscript is well-written. Despite this overall positive evaluation, I have a few comments and concerns which prevent me from recommending publication of the manuscript in its present form.

We thank the reviewer for their overall positive evaluation and for the very helpful comments.

DATE
09 July 2021

PAGE
9/17

- The authors provide a nice overview of the existing literature on reward and positive affect effects on cognitive control. But they missed a few relevant papers by Hefer and Dreisbach:

o Hefer & Dreisbach (2016). The motivational modulation of proactive control in a modified version of the AX-continuous performance task: Evidence from cue-based and prime-based preparation. *Motivation Science*, 2(2), 116-134.

<http://dx.doi.org/10.1037/mot0000034>

o Hefer & Dreisbach (2017). How performance-contingent reward prospect modulates cognitive control: Increased cue maintenance at the cost of decreased flexibility. *JEP: LMC*, 43(10), 1643-1658. <http://dx.doi.org/10.1037/xlm0000397>

o Hefer & Dreisbach (2020). Prospect of performance-contingent reward distorts the action relevance of predictive context information. *JEP: LMC*, 46(2), 380-399.

<http://dx.doi.org/10.1037/xlm0000727>

o Hefer & Dreisbach (2020). The volatile nature of positive affect effects: opposite effects of positive affect and time on task on proactive control. *Psychological research*, 84, 774-783. <https://doi.org/10.1007/s00426-018-1086-4>

- I'd like to highlight the findings from Hefer & Dreisbach's work that seem to be most relevant for the present study: 1) Hefer and Dreisbach found that the increased proactive control by reward comes at a cost of decreased flexibility in terms of delayed adaptation to new reward and task conditions. This seems to be important with respect to the comparisons with the extinction phases in the present study. Furthermore, there might be carry-over effects from the contingent reward phase to the non-contingent phase, if the latter is following the first. 2) Hefer and Dreisbach (as well as Fröber & Dreisbach, 2016) found time-on-task effects in the AX-CPT in terms of increasing proactive control over time. These time-on-task effects interfere with rather subtle

DATE
09 July 2021PAGE
10/17

(compared to reward effects) positive affect effects. Due to these two phenomena, I suggest to conduct an analysis with counterbalance order as an additional between-subjects factor. Collapsing across counterbalance order seems not justified, given the existing literature. Conclusions from the present study might be different when the additional factor is included in analyses.

Thank you for these very interesting suggestions. The evidence for increased proactive control by reward at a cost of decreased flexibility found by Hefer & Dreisbach is particularly interesting in the context of the present study. These results are now included in the discussion of our manuscript (page 8): “Second, we did not find a significant difference between the contingent outcome phase and the following extinction phase. This finding might seem surprising because it is at odds with a rich literature on how instrumental extinction result in the decrement of action execution and performance [61, 62]. However, recent work by Hefer & Dreisbach [63-65] employing a rewarded AX-CPT paradigm shows that reward increased cue maintenance at the cost of decreased flexibility to new task/contingency conditions. Notably, the observations by Hefer & Dreisbach [64, experiment 1] are particularly relevant in that increased cue maintenance persisted in a subsequent block even when reward was no longer available, which mirrors the present extinction phase.”

In addition, the time-on-task effect observed in the above studies is particularly interesting considering the design of our experiment. For completion, we ran a similar analysis but with Order as an additional between-subjects factor. However, the analysis on RTs revealed no significant Phase by Order interaction, or Phase, Order, Trial Type

DATE
09 July 2021PAGE
11/17

interaction. The analysis of error rates revealed a significant Phase by Order interaction, and a significant Phase, Order, and Trial Type interaction, but a Bayesian ANOVA on error rates does not support these interactions.

Here, considering that the majority of our results show no significant effect of Order on performance, and considering a potential lack of power for this analysis (a point suggested by the Editor), we chose to not include this supplementary analysis in our manuscript. However, we completely agree that time-on-task, and more generally order, might be an important factor in the context of learned-positive outcome contingency. This possibility is now included in our discussion (page 8): “It is also possible that the persistence of increased proactive control beyond the reward phase [similar to 64] is one of the reasons for the absence of significant differences between the contingent and the non-contingent outcome phases, along with the contribution of the task structure suggested above. Specifically, for half of the participants, the non-contingent outcome phase preceded the contingent one. It is hence possible that reward-triggered proactive control enhancement (partly) persisted during the non-contingent outcome phase. And related to this, Hefer & Dreisbach [67] found that proactive control increases with time-on-task in an AX-CPT paradigm. Again, this could be another factor contribution to the absence of significant difference between the contingent outcome phase and the extinction 1 or the non-contingent outcome phases. To further illuminate the present observation, an interesting route for future studies may be to focus on how reward/positive affect-based performance modulations persist with varying instrumental contingency and lengths of learning phases. Thus, this would imply to investigate not only the direct effect of positive outcome on performance across different

DATE
09 July 2021

PAGE
12/17

contingencies conditions, but also to investigate the long-term effect of these different contingencies.”

- The study has a complex design and in my opinion, the results section would greatly benefit from a more detailed elaboration on the results. More precisely, the interaction of trial type and phase in RTs should be described in more detail than “there is a simple main effect in AX, BX, and BY, but not in AY”. What exactly is meant by simple main effect here? I would have expected to find single comparisons between phases – separately for trial types – here, and not following the main effect of trial type. Likewise, the trial type x phase interaction in error rates is not described with enough detail.

Thank you for this suggestion. This point was also raised by Reviewer 1.

We agree that the description of our results was lacking detail and we changed this in this revised version of the manuscript. We would like to thank you for this suggestion as these additional analyses allow us to qualify more our results.

Minor comment: Please add to the methods section, which exact stimuli from the NimStim face stimulus set and BOSS data set were used in the study.

The references of the exact stimuli are now added to the manuscript.

Reviewer: 3

Prevel and colleagues present a behavioural study, investigating the effect of implicit

DATE
09 July 2021

PAGE
13/17

instrumental contingency of monetary reward and positive affect on cognitive control using an impressively large number of participants. The authors explored whether participants are able to detect and adjust their responses to these implicit contingencies and tested how far the effect of implicit contingency differed between motivational and emotional outcomes. The authors found that implicit contingencies between responses and both monetary and affective outcomes led to significant performance improvement. Furthermore, both implicit reward domains impacted task performance similarly. This is a well-structured, well-written and methodologically sound behavioural study. The design is well-conceived and the statistical analysis appropriate. I fully recommend publication, however, I do request some additional discussion to widen the scope of this study.

We would like to thank Reviewer 3 for its positive evaluation of our manuscript and for its suggestions.

Specifically, I would appreciate if the authors could add some more discussion regarding the potential underlying neural processes as well as some speculation as to how the results and or simply the interesting paradigm might be useful in the clinical domain.

Recently there have been several studies linking neural mechanisms of arousal and cognitive control to psychopathological disorders such as anxiety, depression or even to relapse after therapy.

Berwian IM, Wenzel JG, Collins AGE, Seifritz E, Stephan KE, Walter H, Huys QJM. Computational Mechanisms of Effort and Reward Decisions in Patients With Depression and Their Association With Relapse After Antidepressant Discontinuation. JAMA Psychiatry. 2020 May 1;77(5):513-522. doi:

DATE
09 July 2021

PAGE
14/17

10.1001/jamapsychiatry.2019.4971. PMID: 32074255; PMCID: PMC7042923.

Grueschow M, Stenz N, Thörn H, Ehlert U, Breckwoldt J, Brodmann Maeder M, Exadaktylos AK, Bingisser R, Ruff CC, Kleim B. Real-world stress resilience is associated with the responsivity of the locus coeruleus. Nat Commun. 2021 Apr 15;12(1):2275. doi: 10.1038/s41467-021-22509-1. PMID: 33859187; PMCID: PMC8050280.

Grueschow M, Kleim B, Ruff CC. Role of the locus coeruleus arousal system in cognitive control. J Neuroendocrinol. 2020 Dec;32(12):e12890. doi: 10.1111/jne.12890. Epub 2020 Aug 20. PMID: 32820571.

Kaldewaij R, Koch SBJ, Hashemi MM, Zhang W, Klumpers F, Roelofs K. Anterior prefrontal brain activity during emotion control predicts resilience to post-traumatic stress symptoms. Nat Hum Behav. 2021 Feb 18. doi: 10.1038/s41562-021-01055-2. Epub ahead of print. PMID: 33603200.

I believe the manuscript could benefit from incorporating these papers and related work to outline potential paths for diagnosis, intervention or treatment related to psychopathological disorders and thereby substantially enhancing the scope of the results.

We agree that a discussion on the potential underlying neural processes and on how the results might be useful in the clinical domain is valuable. These two aspects are now discussed in the final paragraph of the revised manuscript (page 8): “Finally, we will discuss the resent findings in the light of potential underlying neural mechanisms and implications for the clinical domain. Concerning the neural processes, it is commonly established that the prefrontal cortex plays an important role in cognitive control, and notably in the active maintenance of cue information. Studies have

DATE
09 July 2021PAGE
15/17

demonstrated that reward prospect enhances prefrontal cortex activity, which in turn is thought to modulate performance in working memory tasks [71, 72]. Moreover, a vast number of studies have shown that a network of cortical and subcortical regions (including the ventral striatum and the anterior cingulate cortex) are involved in increasing attention and cognitive control in various tasks to maximize reward outcomes [5, 73]. More recently, the locus coeruleus-norepinephrine system has been linked to the allocation of cognitive control [74, 75]. In the context of the present study, it might be interesting to investigate in how far activity modulations in these regions would vary with the degree of contingency between the performed action and the outcome (e.g., with Δ "P"), as well as with the type of outcome (monetary reward versus positive affective stimuli). With regard to clinical implications, the evidence that performance in a cognitive control task is facilitated by positive outcomes, and that these modulations can persist even when the reward is no longer delivered, may be relevant for cognitive control training approaches. For example, the allocation of cognitive effort is reduced in patients with depression [76]. It could be interesting to investigate whether cognitive control training involving reward would increase the allocation of control in these patients, and in which conditions this increment could persist in time even in the absence of additional positive outcomes. That said, it is important to consider that not only the willingness to invest cognitive control is often impaired in depressed patients, but also the ability to process reward signals [77]."

In the introduction the AX-CPT paradigm falls a bit out of the sky and leaves the non-

DATE
09 July 2021PAGE
16/17

expert reader wondering what this is. I believe it would be helpful to add one or two very general and explanatory sentences to alleviate this issue.

Thank you for this suggestion. We agree that the AX-CPT paradigm was maybe not described with enough detail, and it might be difficult to understand the rationale for why we chose this task for a non-expert. In the revised manuscript, we now elaborate on the paradigm (page 1): “For example, evidence was found that reward increases the maintenance of cue information in working memory tasks, which is considered to reflect increased proactive control. Prominent demonstrations involve the AX-CPT paradigm, in which participants have to respond to a probe stimulus (‘X’) by executing a specific target response, but only when the probe is preceded a specific cue (‘A’). Otherwise, participants have to produce a non-target response to less frequent sequences (typically, 70% of AX trials, and 10% of AY, BX, and BY trials each). Increased maintenance of cue information caused by the presentation of reward is measured by global speed-up but decreased accuracy in AY trials [19], the performance decrement in AY trials being interpreted as an increased preparation to the target X from cue A. In contrast to this, embedding pictures or videos with positive or negative content in cognitive control tasks did yield mixed effects on performance. While studies reported more frequently increased cognitive flexibility [12, 20] caused by (positive) affect, with lower error rates on AY trials and/or increased error rates on BX trials, other studies reported no impact of affective manipulations on cognitive control [21-23] or even increased cognitive stability/proactive control [14].”

DATE
09 July 2021

PAGE
17/17

Is it possible that the following sentence: 'To match the contingent condition, positive outcomes were delivered with the same probability, but independent of response speed and accuracy.' Is missing a statement along the lines of 'in the non-contingent condition', just to make this clear?

Thank you for noticing this missing statement. We added “in the non-contingent condition” to the new manuscript.

Page 5 line 56: 'Responses times' should read 'Response times'

We made the correction in the manuscript.